# A global analysis of how human infrastructure squeezes sandy coasts

Eva M. Lansu [1,2] ✉, Valérie C. Reijers [3], Solveig Höfer [1,2], Arjen Luijendijk[4,5], Max Rietkerk[6], Martin J. Wassen [6], Evert Jan Lammerts[7] & Tjisse van der Heide [1,2] ✉

Coastal ecosystems provide vital services, but human disturbance causes massive losses. Remaining ecosystems are squeezed between rising seas and human infrastructure development. While shoreline retreat is intensively studied, coastal congestion through infrastructure remains unquantified. Here we analyse 235,469 transects worldwide to show that infrastructure occurs at a median distance of 392 meter from sandy shorelines. Moreover, we find that 33% of sandy shores harbour less than 100 m of infrastructure-free space, and that 23–30% of this space may be lost by 2100 due to rising sea levels. Further analyses show that population density and gross domestic product explain 35–39% of observed squeeze variation, emphasizing the intensifying pressure imposed as countries develop and populations grow. Encouragingly, we find that nature reserves relieve squeezing by 4–7 times. Yet, at present only 16% of world's sandy shores have a protected status. We therefore advocate the incorporation of nature protection into spatial planning policies.

The Earth's coastal zones exhibit a wide variety of landforms ranging from rocky shores dominated by cliffs and cobblestone beaches to soft-sediment shores typified by tidal flats and sandy beaches[1]. Sandy coasts cover one-third of the world's ice-free shorelines[2], are naturally characterized by gentle sea-to-land gradients, and may extend over many kilometres inland[1]. These naturally long gradients typically provide space to support a sequence of habitats including reefs, seagrasses, sandy beaches, dunes and grasslands, shrublands and forests that are connected through exchange of sediments, nutrients, water, plant and animal species[3–5]. The habitats within the sequence often depend on each other for providing many ecosystem services to humanity, including flood defence, carbon storage, recreation, freshwater storage, and biodiversity[3,6–8].

Although humans increasingly depend on coastal ecosystem services – around 40% of our population lives within 100 km of the shoreline and this number continues to rise[9,10] – human development and exploitation simultaneously cause their rapid degradation[11–15]. At a global scale, sea level rise and an increase in extreme weather events, such as heat waves[16], droughts and precipitation extremes[17,18], threaten coastal ecosystems. These global impacts are augmented by local impacts, such as eutrophication, salinization and pollution[19,20]. However, arguably the most important local disturbance is infrastructure development close to shore[21]. Infrastructure restricts the available space to accommodate coastal ecosystems and impedes cross-ecosystem processes through landscape fragmentation. The space reduction hampers sandy coasts and their habitats to adapt to sea level rise by landward retreat – a phenomenon called 'coastal squeeze'[22,23]. While many recent studies have assessed shoreline changes, highlighting the threat of coastal erosion and sea level rise[24–28], a global assessment of how human infrastructure squeezes coastal ecosystems from the landward side has not yet been performed[29].

[1]Department of Coastal Systems, Royal Netherlands Institute for Sea Research (NIOZ), Den Burg, The Netherlands. [2]Conservation Ecology Group, Groningen Institute for Evolutionary Life Sciences, University of Groningen, Groningen, The Netherlands. [3]Faculty of Geosciences, Department of Physical Geography, Utrecht University, Utrecht, The Netherlands. [4]Department of Resilient Ports and Coasts, Deltares, Delft, The Netherlands. [5]Department of Hydraulic Engineering, Faculty of Civil Engineering and Geosciences, Delft University of Technology, Delft, The Netherlands. [6]Copernicus Institute of Sustainable Development, Environmental Sciences Group, Utrecht University, Utrecht, The Netherlands. [7]Programma Deltanatuur, Staatsbosbeheer, Amersfoort, The Netherlands. ✉e-mail: eva.lansu@nioz.nl; tjisse.van.der.heide@nioz.nl

Here, we analysed the proximity of human infrastructure to the world's sandy shores, which are typified by loose deposits of sand with occasional gravel and shells[2]. Using a recent global dataset of sandy shores[2], we constructed 235,469 25-km long transects perpendicular to the shoreline at 1-km interspaces, combined representing 29% of the world's ice-free shoreline. Next, we used data from OpenStreetMap and Global Urban Footprint[30] to calculate where the nearest paved road or building intersected each transect, defining the infrastructure-free width as the distance between the shore and the first structure. We also identified which transects already crossed natural cliff obstructions or intersections with the shoreline (i.e., on narrow land strips) prior to intersecting infrastructure. Next, we explored to what degree basic socio-economic variables can explain infrastructural squeeze and investigated if nature reserves[31] can successfully relieve squeezing. Finally, we calculated the percentage of remaining infrastructure-free space under various sea level rise projections.

## Results and discussion

### The current state of coastal squeeze by infrastructure

We found human structures to be ubiquitous along sandy shores worldwide. We found that of the 235,469 transect analysed, 28% were limited by natural barriers caused by coastal geometry or steep cliffs before being intersected by any paved roads or buildings (Supplementary Fig. 2). Of the remaining 168,654 transects, 93% were confined by buildings and/or paved roads within the first 25 km of the coastal zone. The distribution of the infrastructure-free width is overall positively skewed: the nearest structure is mostly found at a short distance (Supplementary Fig. 3)[32]. The median distance between the shoreline and the nearest structure is only 392 m. Moreover, 33% of global sandy shores harbour less than 100 m infrastructure-free space, implying infrastructure development directly on or near the beach. When only heavy infrastructure – i.e., buildings and highways (Supplementary Fig. 1)[32] – is included, median width increases to 1.6 km, with 28% of the shores having such structures within the first 100 m from the waterline. These are conservative estimates as our dataset does not include areas where the beach has entirely disappeared due to construction of dikes and dams.

Infrastructure is generally closer to sandy shores in densely populated areas, particularly between 32 and 45 degrees North (Fig. 1 & Supplementary Fig. 4). Shores in this latitudinal band have a median infrastructure-free width of 70 m. This band includes Japan, South Korea, Lebanon, Syria, Turkey, Italy, France, Spain, and the United States of America. All these countries rank in the top 20 of most severely squeezed countries in the world (Supplementary Fig. 5). At the continental level (Fig. 2a), infrastructure-mediated coastal squeeze is most severe in Europe (median: 131 m), followed by Asia (151 m), North America (402 m) and South America (764 m). Sandy coasts of Africa (1.6 km) and Oceania (2.8 km) are less confined.

### Socio-economic drivers and nature protection

To explore how socio-economic factors affect coastal squeeze by infrastructure, we constructed a simple multiple regression model using two basic proxies: coastal population density and gross domestic product (GDP). The resulting model explains 35% of the observed variance in country medians for infrastructure-free width (Supplementary Table 1a). Population density and GDP both reduce the infrastructure-free width, with the effect size of population density being slightly larger than that of GDP. The model's explanatory potential increases to 39%, when only heavy infrastructure is considered (Supplementary Table 1b). Analyses of nature reserves versus unprotected shores highlight that protected shores have a four times greater infrastructure-free coastal width (1.4 km) compared to unprotected sandy shores (302 m) (Fig. 2b). This is even clearer when only heavy infrastructure is considered, as the width of protected shores (8.2 km) becomes seven times greater than of non-protected shores (1.1 km) (Supplementary Fig. 6d). Further analyses reveal that most of the protected areas are situated in rural (95%) rather than in urban areas (5%). Moreover, we find that protected areas have a 3.0 times larger infrastructure-free zone compared to non-protected rural areas, while this difference equals only 1.7 in urban areas. Although correlative, these findings suggest that the anthropogenic pressure in urban coastal areas may prohibit the creation of protected areas, possibly limiting their potential to preserve the unimpacted space between the shoreline and infrastructure.

### Accommodation space for coastal retreat

Overall, we demonstrate that the world's sandy coasts are severely squeezed by human-made infrastructure. In line with earlier work[33],

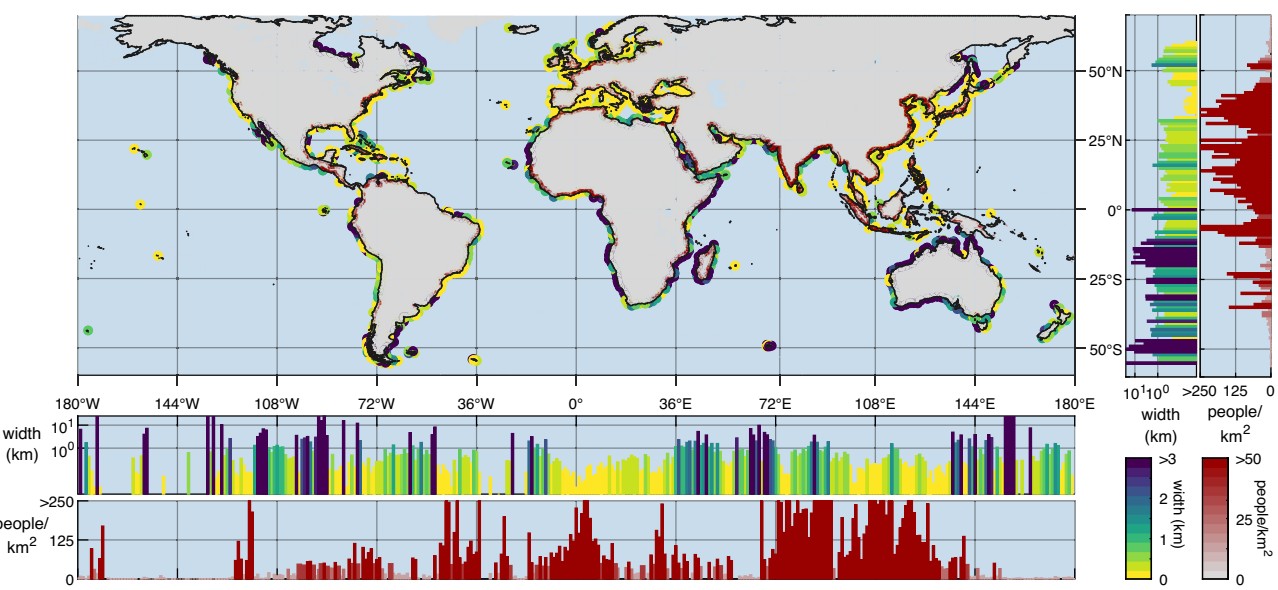

**Fig. 1 | Map of the coastal squeeze by human infrastructure along the world's sandy shores.** Infrastructure-free width is depicted in yellow-green-blue from 0 to 3 km. Coastal population density (data obtained from WorldPop[64]) is depicted in white-red from 0 to 50 people/km². Bar graphs show the latitudinal and longitudinal average. The figure was created using the Matlab mapping toolbox. Source data are provided as a Source Data file[66].

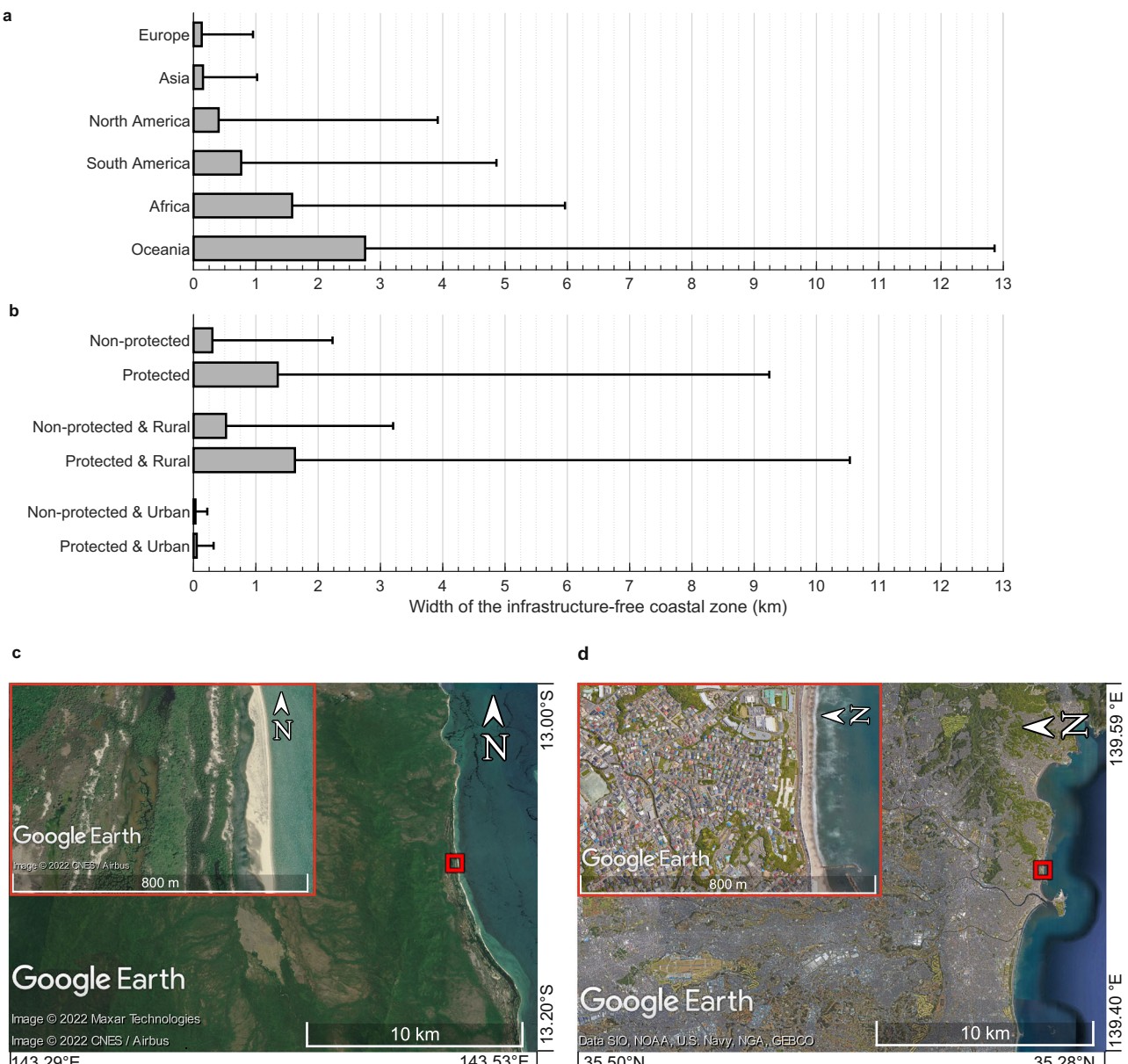

**Fig. 2 | Infrastructure-mediated coastal squeeze by continent and inside/outside nature reserves.** Panel (**a**) depicts the infrastructure-free width per continent and (**b**) shows nature protected areas (16% of world's sandy shore) versus non-protected areas for all data combined (Wilcoxon rank-sum test: Z = 71.6, $p < 0.001$) and split between urban and rural areas. Urban (non) protected and rural (non) protected groups have significantly different distributions (Kruskal-Wallis test: $\chi^2 = 22189$, $p < 0.001$, df = 3). Additional posthoc analysis demonstrated that all groups significantly differed from each other. Bars represent medians and whiskers 75th percentiles. Panel (**c**) and (**d**) highlight typical examples of infrastructure-free versus highly squeezed coastal areas. Map data @2022 Google. Source data are provided as a Source Data file[66].

our findings suggest that demand and budget for infrastructure are closely related to coastal squeeze intensity. Because both coastal population density[9,10] and GDP[34] are projected to grow over the coming decades, infrastructure development close to shore is also expected to increase. The continued encroachment of infrastructure on the shoreline has major consequences for the adaptive capacity of sandy coasts as human infrastructure can hamper sediment transport and supply[35,36] and impede landward migration[22,37,38]. While climate change projections predict continued sea level rise globally[18,39], consequences for sandy shores locally differ due to variations in coastal geomorphology, sediment availability and the rate of sea level rise[25,40,41]. Despite opposing views on the use of global-scale assessments due to their uncertainties, both sides stress that infrastructure poses the biggest threat to shoreline migration, and explicitly mention the lack

of a global dataset to account for accommodation space[25,40,41]. To provide a first estimate on where coastal squeeze may pose future problems, we used the only global assessment of future shoreline change available to date[25]. Our calculations show that projected retreats by Vousdoukas et al. (2020) exceed the current infrastructure-free width along 23% of the world's sandy shores for Representative Concentration Pathway (RCP) 4.5 and 30% for RPC 8.5 by 2100[32]. More specifically, shoreline retreat could drown remaining infrastructure-free zones along 31% of European, 28% of Asian, 22% of North American, 18% of South American, 12% of Oceanian and 11% of African shores under RPC 4.5, and a larger proportion under RPC 8.5 (Supplementary Table 2). Clearly, these estimates are crude as they do not consider uncertainties in beach behaviour under rising sea levels, nor the available beach-dune sediment budgets[40]. Nevertheless, they do

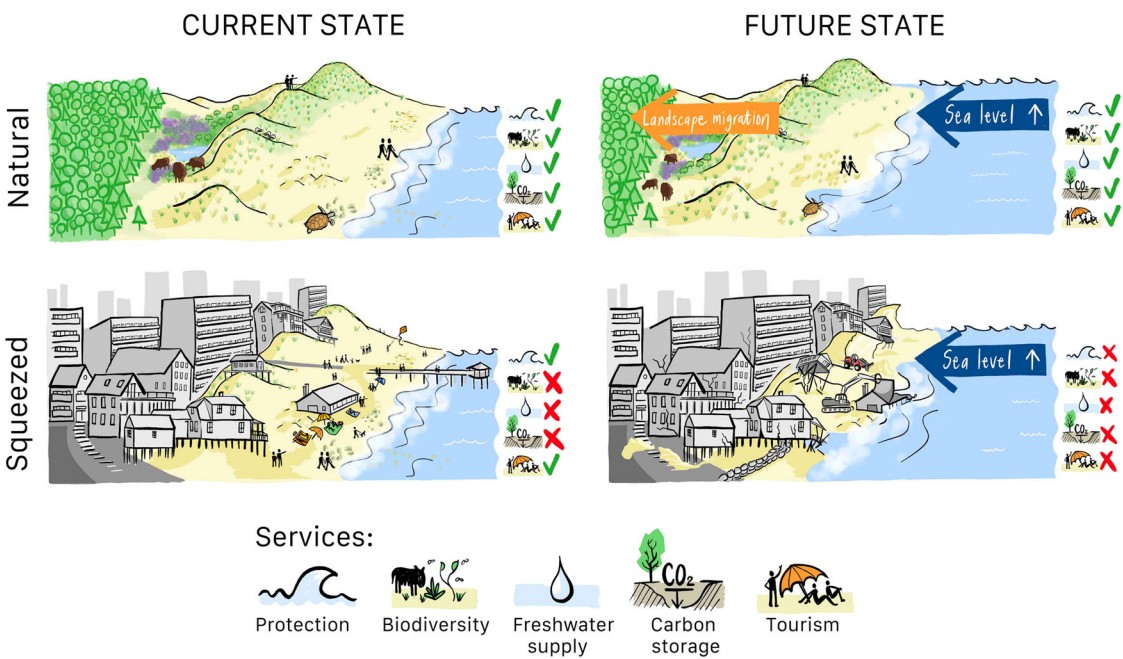

**Fig. 3 | Consequences of coastal squeeze in current and future conditions.** Whereas the natural sandy coast migrates in response to coastal retreat, the squeezed coast, and its services collapse.

illustrate that coastal squeeze will increasingly pressure the world's sandy shores in the foreseeable future.

## Implications for ecosystem functioning

Infrastructure development close to shore also has major implications for ecosystem functioning (Fig. 3). Ecological theory predicts that 50% of plant and animal species are lost by an order-of-magnitude decline in habitat size[42]. In coastal dunes of the Mediterranean and the Gulf of Mexico, dramatic habitat and species losses have been attributed to infrastructure development[12,43]. Loss of pristine, infrastructure-free coastal zones is likely an important factor in such observed degradation. Sufficiently wide coastal zones are vital for preserving natural succession-disturbance dynamics that generate high habitat diversity, which in turn supports a diverse assembly of unique plant and animal species[44–46]. In addition, lack of space also threatens natural coastal defence functions. Previous studies found that beaches should typically be over 300 m wide to effectively reduce erosion and support natural dune development[47–49]. Our results, however, highlight that 46% of the world's sandy shores currently have less than 300 m of infrastructure-free space, implying that such areas could be at risk for erosion. As a consequence, ecosystem services such as biodiversity, carbon storage, recreation, freshwater supply, and flood defence are all under threat[50].

## Measures for the future

Our global analysis can aid policy makers to design appropriate regional measures in response to coastal squeeze. The present default policy is often aimed at keeping the shoreline in place by applying hard-structure engineering or beach nourishments[51,52]. An alternative strategy, seaward movement of the shoreline through mega-nourishments, is expensive and requires high sediment availability[50,51], but can be preferred in hyperdeveloped coastal zones. However, when accommodation space is sufficient, we suggest that *managed retreat*[53–55] has the highest potential to preserve coastal resilience by allowing retreat and sediment-exchange with the backshore. However, the accommodation space is often lacking. As highlighted by the relatively small differences in infrastructure-free space between protected and non-protected areas in urban

regions, nature reserves are not a silver bullet for relieving coastal squeeze. Nevertheless, our correlative analyses do suggest that they can be effective in preserving the remaining accommodation space, as they on average support a two times greater infrastructure-free coastal width. Yet only 16% of world's sandy shore currently has a protected status. Meanwhile, services provided by intact, well-functioning coastal ecosystems are gaining importance as coastal populations continue to grow[9,10]. We suggest that coastal nature reserves can play an important role in preserving ecosystem functioning and coastal resilience to sea level rise. Therefore, we argue that protection of intact infrastructure-free coasts should be integrated in spatial planning policies.

## Methods

### Computation infrastructural squeeze

To obtain the most accurate results from our analyses, we used the latest publicly available dataset with the greatest detail for each data layer (Supplementary Table 3). We started by drawing 25 km long transects perpendicular to the coastline with an interspacing of 1 km. We used the coastline of OpenStreetMap, which indicates the mean high-water springs line at a resolution comparable to shoreline estimates derived from Sentinel-1 with a 10-m spatial resolution[56–58]. We then selected the transects where the coast was classified as sandy[2]. Next, we downloaded paved streets from OpenStreetMap and building data from Global Urban Footprint which together provide the greatest spatial detail with regard to infrastructure of publicly available datasets[59,60]. Next, we calculated two proxies to assess the degree of coastal squeeze by human infrastructure per transect. The first proxy is the *infrastructure-free zone*, which is defined as the distance between the shore and any hard human-made structure, including: buildings, freeways, car roads, paved bike and pedestrian paths (Supplementary Fig. 1). The second proxy is the *heavy infrastructure-free zone*, which we defined as the distance between the shore and the first building or freeway, thus ignoring smaller car roads, bike and pedestrian paths. As these proxies estimate the proximity of paved roads and buildings, unpaved seawalls and dikes are not considered. Next, we identified transects where natural obstructions from cliffs or intersections

with the shoreline (i.e., on narrow land strips) occurred prior to intersecting infrastructure. We used the CoastalDEM to identify steep slopes or cliffs, which is an improved version of SRTM and currently the most accurate elevation map for coastal regions[61]. As this coastal terrain model is limited to latitudes below 60°N, we excluded 7% of our transects beyond this northern limit. We then calculated the maximum seaward slope along each transect, and excluded transects with slopes steeper than the angle of repose of dry sand of 34°[62], thus conservatively assuming that steeper slopes are not sandy and thus prevent further landward retreat. We found that 4% of the sandy shores have a steep slope within the infrastructure-free zone and 5% within the heavy infrastructure-free zone. Finally, we identified any transects, that started on land, but intersected the shoreline prior to intersecting any human-made structure. For instance, transects can be short because they are located on a small island or peninsula, and not because of squeeze by infrastructure. Supplementary Fig. 2 highlights which transects were excluded based on the Northern limits of the CoastalDEM, and which were limited by natural obstructions rather than human-made infrastructure. The resulting frequency distributions of infrastructure-free and heavy infrastructure-free coastal widths for the remaining transects squeezed by human infrastructure are presented in Supplementary Fig. 3.

### Regression model
We constructed a simple multiple linear regression model with coastal population density and gross domestic product per capita (GPD) as explanatory variables to explore how basic socio-economic factors affect coastal squeeze by infrastructure. Data on GDP was available on country scale, and population density was available as 1 km-resolution, georeferenced images. We selected Worldpop as a source for coastal population density, because of its high accuracy, data accessibility and consistency[63,64]. We sampled population density along each full transect (~25 km long) and computed median values per country. To best approximate a normal distribution of residuals, we log-transformed both explanatory variables. The correlation among both explanatory variables was weak ($R = -0.22$).

### Protected coastal areas
We computed the infrastructural squeeze for transects with and without a protected status (World Database on Protected Areas). For our analyses, we considered a transect as protected when it was located within a protected area in the first two kilometers from the sandy shore. Next, we statistically compared the protected versus non-protected for both infrastructure-free and heavy infrastructure-free coastal widths using Wilcoxon rank sum tests. As a second step, we explored how protected and non-protected coastal widths compare between urban and rural areas. To distinguish between rural and urban areas, we used a threshold of 300 people/km$^2$, following the urban cluster definition of the European Committee[65]. Next, we statistically compared the coastal widths by applying a Kruskal-Wallis test, followed by a posthoc analysis based on Wilcoxon's rank-sum test with a Bonferroni correction of the significance levels.

### Projected shoreline retreat
We compared the infrastructure-free space with projected long-term shoreline change[25], resulting from sea level rise based on the 50th percentile under both RCP 4.5 and 8.5. In 95 percent of cases, we found a shoreline projection within 0.05 degrees (~5.6 km) of our transects. Next, we subtracted this reported shoreline retreat from the infrastructure-free coastal width. An outcome equal or smaller than 0 kilometres means that (more than) the infrastructure-free space would be taken by the projected retreat of 2100. We computed both globally and per continent what percentage of transects would completely lose their infrastructure-free space.

## Data availability
The computed (heavy) infrastructure-free coastal widths are provided in the Source Data file. The data underlying Fig. 1, Fig. 2 and Supplementary Figs. 2–6, and Supplementary Table 1 and 2 are also provided in the Source Data file, which is deposited in Zenodo (https://doi.org/10.5281/zenodo.7525228). All other relevant data is available upon request. Source data are provided with this paper.

## Code availability
Script analyses are deposited in Zenodo: https://doi.org/10.5281/zenodo.7525228.

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

## Acknowledgements

We thank Rogier Schouten for his help with the data analysis, Elco van Staveren (Thinksketch.com) for designing Fig. 3, and Christine Angelini for

her comments on a previous version of the manuscript. EL was funded by NWO-LLDD grant 17595; SH by OBN grant OBN-2019-105-DK; VR by NWO-Veni grant VI.Veni.212.059; MR by NWO grant OCENW.M20.169 and ERC-Synergy grant 101071417; TH was supported by NWO/TTW-Vidi grant 16588.

## Author contributions

E.L., T.H. and V.R. conceptualized the study; E.L., A.L., T.H. and V.R. designed the methodological approach; E.L. performed the analyses and prepared the visual materials; E.L., T.H. and V.R. wrote the original draft; E.L., V.R., S.H., A.L., M.R., M.W., E.J.L., T.H. contributed to further writing and editing of the paper.

## Competing interests

The authors declare no competing interests.
