## [Peer Review File · Nature Communications]

A global analysis of how human infrastructure squeezes sandy
coastsReviewer #1 (Remarks to the Author):

I have now read the revised manuscript and the response to the reviewers' comments. The authors have addressed my comments on the previous version in a comprehensive manner. The additional analysis has also strengthened the manuscript.

I still have two further minor comments/questions that the authors should consider:

1. The justification on the use of datasets is not complete and partly not correct - possibly the authors refer to resolution and not accuracy (for the GUF data?) while for the population dataset they mention accuracy but only cite a study for China (and not global). Further, no justification is included for the use of CoastalDEM. The selection of datasets is important, not only for understanding and contextualising the results but also because they are informative for other researchers who need to make similar decisions regarding the use of global datasets.

2. Lines 123-125 are essentially repeating lines 103-105, or am I missing something?

Overall, I find the revised paper interesting and a useful first attempt at quantifying the global effects of coastal squeeze due to the presence/development of coastal infrastructure.

Reviewer #2 (Remarks to the Author):

Very thorough responses, from my point of view, to all of the three Reviewer reports and I have nothing more to ask for. I am sure the the paper will elicit good interest.

Very minor edits that may be rectified at the proof stage:

- Lines 41-42: "Sandy coasts cover one-third of the world's ice-free shoreline and are naturally characterized by gentle sea-to-land gradients that extend over many kilometres inland". I suggest a more nuanced statement as not all of the sandy coasts extend over many km inland and the word extension here refers to the gradient, so the statement is syntactically awkward. I suggest this or something similar: "Sandy coasts cover one-third of the world's ice-free shorelines, are naturally characterized by gentle sea-to-land gradients, and may extend over many kilometres inland".

- Lines 79-82: "... 239,966 transects analysed, 38% were limited ... 95% were confined"

- Line 117: "These findings suggest ..."

- Line 141: "More specifically, shoreline retreat 'could drown' (in lieu of 'will drown')."

Edward Anthony, Aix en Provence, France.

Reviewer #3 (Remarks to the Author):

This is a resubmission, and this reviewer was not one of the original reviewers. Having reviewed the original set of referee comments and the author responses thereto, however, it is clear that the comments were addressed.

This is an important analysis of the type and density of infrastructure development impacting the global coastline and merits publication after an additional clarification. What is unacceptably lacking is a broader discussion on global coastline data, both sources and spatial resolution. There are at least two public domain global shoreline vector resources that are newer (e.g. Sayre et al., Liu et al.) much better quality, and at a much higher spatial resolution than the datalayer the authors chose to use (GSHHS). Therefore, their reasoning for choosing to use the GSHHS dataset needs to be included, and not acknowledging the existence of higher resolution global shoreline

resources would suggest an unfamiliarity with the literature. In addition to the justification of source dataset, the authors should include a short discussion on how scale matters in their analysis, and in particular how the results might differ depending on the resolution of the global shoreline vector (e.g. 30 m vs 250 m vs 1000m).

Otherwise, this is an important publication which will help to mainstream the concept of coastal squeeze and could improve coastal planning in general and siting of infrastructure development in particular.

Dear editor and reviewers,

We wish to thank the reviewers for their constructive and helpful comments on our manuscript entitled: *"A global analysis of how human infrastructure squeezes sandy coasts"* (#NCOMMS-23-22848-T). Below, we present a detailed point-by-point reply on how we dealt with all comments.

We hope that we attended to all your comments and suggestions to your satisfaction.
Sincerely,

Tjisse van der Heide and co-authors

Reviewer #1

I have now read the revised manuscript and the response to the reviewers' comments. The authors have addressed my comments on the previous version in a comprehensive manner. The additional analysis has also strengthened the manuscript.

I still have two further minor comments/questions that the authors should consider:

1. The justification on the use of datasets is not complete and partly not correct - possibly the authors refer to resolution and not accuracy (for the GUF data?) while for the population dataset they mention accuracy but only cite a study for China (and not global). Further, no justification is included for the use of CoastalDEM. The selection of datasets is important, not only for understanding and contextualising the results but also because they are informative for other researchers who need to make similar decisions regarding the use of global datasets.

Reply: We agree that the criteria for selecting the datasets should be clearly stated. We now start the Methods description by stating that we only used publicly available data and refer immediately to Supplementary Table 3 that provides an overview (lines 336-337): *"To obtain the most accurate results from our analyses, we used the latest publicly available dataset with the greatest detail for each data layer (Supplementary Table 3)."*

Furthermore, as, requested, we provide justifications for use of OSM and GUF data. We now refer to studies that assess the data accuracy and completeness on a global scale (lines 341-344): *"Next, we downloaded paved streets from Open Street Map and building data from Global Urban Footprint which combined provide the greatest spatial detail with regard to infrastructure of publicly available datasets^{59,60}."*

Worldpop data (ln 375-376): *"We selected Worldpop as a source for coastal population density, because of its high accuracy, data accessibility and consistency^{63,64}."*

CoastalDEM (ln 353-355): *"We used the CoastalDEM to identify steep slopes or cliffs, which is an improved version of SRTM and currently the most accurate elevation map for coastal regions⁶¹."*

References:

59. Barrington-Leig et al. *The world's user-generated road map is more than 80% complete*. *PLoS One* **12**, 1–20 (2017).
60. Esch et al. *Where we live-A summary of the achievements and planned evolution of the global urban footprint*. *Remote Sens.* **10**, (2018).
61. Kulp et al. *Climate Central Scientific Report CoastalDEM v2.1: A high-accuracy and high-resolution global coastal elevation model trained on ICESat-2 satellite lidar*. 1–17 (2021).
63. Lloyd, C. T. et al. *Global spatio-temporally harmonised datasets for producing high-resolution gridded population distribution datasets*. *Big Earth Data* **3**, 108–139 (2019).
64. Tatem. *WorldPop, open data for spatial demography*. *Sci. Data* **4**, 2–5 (2017).

2. Lines 123-125 are essentially repeating lines 103-105, or am I missing something?

Reply: We removed this repetition by deleting lines 123-125: *“Two basic proxies for socio-economy – population density and GDP – statistically explain 40-43% of variance.”*

Overall, I find the revised paper interesting and a useful first attempt at quantifying the global effects of coastal squeeze due to the presence/development of coastal infrastructure.

Reply: Thank you for reviewing our manuscript again.

Reviewer #2

Very thorough responses, from my point of view, to all of the three Reviewer reports and I have nothing more to ask for. I am sure the the paper will elicit good interest.

Very minor edits that may be rectified at the proof stage:

3. Lines 41-42: "Sandy coasts cover one-third of the world's ice-free shoreline and are naturally characterized by gentle sea-to-land gradients that extend over many kilometres inland". I suggest a more nuanced statement as not all of the sandy coasts extend over many km inland and the word extension here refers to the gradient, so the statement is syntactically awkward. I suggest this or something similar: "Sandy coasts cover one-third of the world's ice-free shorelines, are naturally characterized by gentle sea-to-land gradients, and may extend over many kilometres inland".

Reply: We agree and adjusted the sentence (lines 41-42): *“Sandy coasts cover one-third of the world’s ice-free shorelines⁸, are naturally characterized by gentle sea-to-land gradients, and may extend over many kilometres inland⁷.”*

4. Lines 79-82: "... 239,966 transects analysed, 38% were limited ... 95% were confined"

Reply: Adjusted.

5. Line 117: "These findings suggest ..."

Reply: Adjusted.

6. Line 141: "More specifically, shoreline retreat 'could drown' (in lieu of 'will drown')."

Reply: Adjusted.

Edward Anthony, Aix en Provence, France.

Thank you for your constructive review.

Reviewer #3

This is a resubmission, and this reviewer was not one of the original reviewers. Having reviewed the original set of referee comments and the author responses thereto, however, it is clear that the comments were addressed.

7. This is an important analysis of the type and density of infrastructure development impacting the global coastline and merits publication after an additional clarification. What is unacceptably lacking is a broader discussion on global coastline data, both sources and spatial resolution. There are at least

two public domain global shoreline vector resources that are newer (e.g. Sayre et al., Liu et al.) much better quality, and at a much higher spatial resolution than the datalayer the authors chose to use (GSHHS). Therefore, their reasoning for choosing to use the GSHHS dataset needs to be included, and not acknowledging the existence of higher resolution global shoreline resources would suggest an unfamiliarity with the literature.

Reply: In hindsight, we agree with the reviewer that we could have selected a more recent and detailed shoreline map for our analyses. Therefore, we have now replaced the GSHHS shoreline with the most recent, more detailed OpenStreetMap shoreline which has an accuracy comparable to shoreline estimates derived from Sentinel-1 with a 10-m spatial resolution (Chen et al., 2023; Mao et al., 2022; Pelich et al., 2021). Moreover, the OSM shoreline map was also used as a basis by Luijendijk et al (2018), whose data we rely on to identify sandy shores worldwide. Thus, using OSM data results in a more coherent and consistent analyses overall. The re-analysis based on OSM data yields a slight reduction in the number of included transects (from 239,966 to 235,469) and minor shifts in the statistics. For instance, the median distance to the nearest structure decreases from 401 to 392 meters. However, overall trends and conclusions are unaffected. Furthermore, as also requested by reviewer 1, we have updated our justification of the datasets in the manuscript.

8. In addition to the justification of source dataset, the authors should include a short discussion on how scale matters in their analysis, and in particular how the results might differ depending on the resolution of the global shoreline vector (e.g. 30 m vs 250 m vs 1000m).

Reply: We agree that the shoreline resolution should be discussed; the methods now include the sentence (ln 338-341): *“We used the coastline of OpenStreetMap, which indicates the mean high-water springs line at a resolution comparable to shoreline estimates derived from Sentinel-1 with a 10-m spatial resolution⁵⁶⁻⁵⁸.”*

By replacing the GSHHS with the OSM shoreline, we increased the resolution of the shoreline (Chen et al., 2023; Mao et al., 2022; Pelich et al., 2021). Implementing the more detailed OSM shoreline resulted in an increase in variance of the infrastructure width. Also, the number of short infrastructure-free widths slightly increased compared to the GHSSH-based analyses. However, we feel that including such a comparison between shoreline vectors is beyond the scope of the paper as, from a more general perspective, the quality of each data layer affects the outcomes of our analyses to some degree. To clarify this, we now highlight the importance of data layer quality in the first sentence of the methods (lines 336-337): *“To obtain the most accurate results from our analyses, we used the latest publicly available dataset with the greatest detail for each data layer (Supplementary Table 3).”*

References

56. Chen et al. Quality Assessment of Global Ocean Island Datasets. *ISPRS Int. J. Geo-Information* **12**, (2023).
57. Pelich et al. Coastline detection based on sentinel-1 time series for ship- And flood-monitoring applications. *IEEE Geosci. Remote Sens. Lett.* **18**, 1771–1775 (2021).
58. Mao et al. Global coastal geomorphology – integrating earth observation and geospatial data. *Remote Sens. Environ.* **278**, 113082 (2022).

Otherwise, this is an important publication which will help to mainstream the concept of coastal squeeze and could improve coastal planning in general and siting of infrastructure development in particular.

Thank you for your review.

Reviewer #1 (Remarks to the Author):

The responses of the authors to my last two comments are satisfactory and comprehensive. There are therefore no more comments from my side.

Reviewer #3 (Remarks to the Author):

The authors have addressed the question about why they did not use a higher resolution shoreline dataset and instead of defending their original selection of a coarser resource, they redid the entire analysis using a better global shoreline. That is truly to be commended. I feel this work should be published and anticipate it will be an important resource for both the scientific community and the coastal management community.